# Autoantibodies Targeting G-Protein-Coupled Receptors: Pathogenetic, Clinical and Therapeutic Implications in Systemic Sclerosis

**DOI:** 10.3390/ijms25042299

**Published:** 2024-02-15

**Authors:** Marco Binda, Beatrice Moccaldi, Giovanni Civieri, Anna Cuberli, Andrea Doria, Francesco Tona, Elisabetta Zanatta

**Affiliations:** 1Rheumatology Unit, Department of Medicine-DIMED, Padova University Hospital, 35128 Padova, Italy; marco.binda@studenti.unipd.it (M.B.);; 2Department of Cardiac, Thoracic, Vascular Sciences and Public Health, University of Padova, 35128 Padova, Italy

**Keywords:** systemic sclerosis, G-protein-coupled receptors, functional autoantibodies, organ involvement

## Abstract

Systemic sclerosis (SSc) is a multifaceted connective tissue disease whose aetiology remains largely unknown. Autoimmunity is thought to play a pivotal role in the development of the disease, but the direct pathogenic role of SSc-specific autoantibodies remains to be established. The recent discovery of functional antibodies targeting G-protein-coupled receptors (GPCRs), whose presence has been demonstrated in different autoimmune conditions, has shed some light on SSc pathogenesis. These antibodies bind to GPCRs expressed on immune and non-immune cells as their endogenous ligands, exerting either a stimulatory or inhibitory effect on corresponding intracellular pathways. Growing evidence suggests that, in SSc, the presence of anti-GPCRs antibodies correlates with specific clinical manifestations. Autoantibodies targeting endothelin receptor type A (ETAR) and angiotensin type 1 receptor (AT1R) are associated with severe vasculopathic SSc-related manifestations, while anti-C-X-C motif chemokine receptors (CXCR) antibodies seem to be predictive of interstitial lung involvement; anti-muscarinic-3 acetylcholine receptor (M3R) antibodies have been found in patients with severe gastrointestinal involvement and anti-protease-activated receptor 1 (PAR1) antibodies have been detected in patients experiencing scleroderma renal crisis. This review aims to clarify the potential pathogenetic significance of GPCR-targeting autoantibodies in SSc, focusing on their associations with the different clinical manifestations of scleroderma. An extensive examination of functional autoimmunity targeting GPCRs might provide valuable insights into the underlying pathogenetic mechanisms of SSc, thus enabling the development of novel therapeutic strategies tailored to target GPCR-mediated pathways.

## 1. Introduction

Systemic sclerosis (SSc) is a complex autoimmune disease characterised by vasculopathy and immune dysregulation, ultimately resulting in widespread fibrosis of the skin and internal organs. SSc carries considerable disease-related morbidity and mortality [1]. The current worldwide prevalence of SSc is reported to be around 17.6 cases per 100,000, with an annual incidence of 1.4 in 100,000 persons each year [2].

While the *primum movens* triggering SSc remains unclear, the interplay between genetic predisposition and environmental factors, such as viral infections or other pathogens, likely significantly contributes to disease onset [3]. Typically, the initial phase involves microvascular injury, activating endothelial cells (ECs) and causing vascular damage. This initiates an inflammatory response with the production of antibodies targeting various antigens, including G-protein-coupled receptors (GPCRs), and the infiltration of immune cells (e.g., T and B cells) into the damaged tissue. This cascade may lead to chronic vasculopathy and fibrosis [4].

The primary clinical manifestations of SSc predominantly revolve around functional and then progressive occlusive peripheral vasculopathy, detectable in nearly all SSc patients. Besides Raynaud’s phenomenon (RP), vascular abnormalities expand during the disease course, potentially resulting in digital ulcers (DUs), pulmonary arterial hypertension (PAH), and, less frequently, scleroderma renal crisis (SRC) [5,6,7]. Moreover, there is robust evidence that even primary heart involvement (PHI) shares a microvascular origin in SSc [8,9]. Another defining trait of SSc is fibrosis, which manifests variably in the skin and internal organs and may cause severe morbidity and premature death. The extent of fibrosis varies among individuals, accounting for the heterogeneity in SSc clinical presentation [4].

Although autoantibodies are found in more than 95% of patients, the potential pathogenic role of SSc-specific antibodies (i.e., anti-topoisomerase I, anticentromere, and anti-RNA polymerase III) in the development of the disease remains possible but has not yet been conclusively demonstrated [10,11]. In recent years, there has been growing interest in the role of functional antibodies targeting GPCRs in the pathogenesis of autoimmune and/or cardiovascular disease [12]. Among the anti-GPCRs, anti-angiotensin type-1 receptor (anti-AT1R) antibodies have been found in severe vasculopathies associated with malignant hypertension, renal vascular disease, and in women with preeclampsia. There is much evidence in the literature indicating that antibodies against endothelin type A receptor (anti-ETAR) may play a role in the pathogenesis of PAH and dilated cardiomyopathy [13,14,15,16]. The obliterative vascular lesions that occur under these conditions resemble those observed in SSc. In addition, anti-AT1R and anti-ETAR are considered the main non-HLA antibodies involved in allograft transplant rejection, which is an intriguing point considering that graft versus host disease (GVHD) shares several similarities with scleroderma [17]. Altogether, this evidence supports a direct role of both anti-AT1R and anti-ETAR antibodies in inducing vasculopathy and autoimmune dysregulation. This review delves into the potential role of GPCR-targeting autoantibodies in disease pathogenesis, clinical manifestations, and therapeutic options in SSc.

## 2. Overview of Immune Abnormalities, Vasculopathy, and Fibrosis in SSc

### 2.1. Immune Dysregulation and Autoantibodies

The immune system dysfunction in SSc involves both innate and adaptive immune responses. The dysregulation of immune cells, particularly T and B lymphocytes, along with aberrant cytokine signalling, play a pivotal role. T cells, especially CD4+ and CD8+ subsets, infiltrate affected tissues and prompt the release of pro-inflammatory cytokines like interleukin-6 (IL-6) and tumour necrosis factor-alpha (TNF-α), thus contributing to tissue damage and fibrosis. B cells, on the other hand, represent the minority of cells in the inflammatory infiltrates in SSc skin [18]. Nevertheless, the presence of autoantibodies in sera from nearly all patients with SSc, and the effectiveness of anti-B lymphocyte therapies (e.g., Rituximab), suggest that B cells are intimately involved in the pathogenesis of the disease [19,20]. Most B cells found in SSc skin display the activation marker CD19, a cell-surface signal transduction molecule and the most potent positive regulator of B cell activity. The state of enhanced intrinsic B cell activation contributes to the loss of immunologic tolerance and the formation of autoantibodies [3].

A major finding that corroborates the pathophysiological role of B cells is the presence of anti-nuclear antibodies (ANA) and other autoantibodies in patients with SSc [19]. While most autoantibodies detected to date have not yet been linked to the molecular pathogenesis of SSc [10,11], some recently identified autoantibodies appear to directly contribute to the development of the disease (i.e., functional antibodies) [16]. Among functional antibodies, besides those targeting GPCRs, anti-endothelial cell autoantibodies (AECA) are a heterogeneous group that can bind to proteins and molecules expressed on the ECs’ surface, resulting in several pathophysiological effects, such as direct or indirect cytotoxicity, ECs apoptosis and activation with increased leukocyte adhesion, coagulation activation, vascular thrombosis, and the release of profibrotic (transforming growth factor-beta, TGF-β) and vasoactive (endothelin-1, ET-1) mediators [21]. Recent evidence suggests that AECA may act as main players in early-stage PAH, DUs, and the peripheral damage detected by nailfold videocapillaroscopy, highlighting the connection between autoimmunity and endothelial damage [22,23]. A pathophysiological role may also be attributed to stimulatory autoantibodies against the platelet-derived growth factor receptor (PDGFR), which contributes to the pathogenesis of SSc by triggering the proliferation of fibroblasts and smooth muscle cells via two tyrosine-kinase receptors, PDGFRα and PDGFRβ [24]. Although no clear associations have been reported between anti-PDGFR and clinical features in SSc patients, a recent study found that agonistic antibodies prompted the proliferation and migration of human pulmonary vascular smooth muscle cells in vitro; this suggests that anti-PDGFR antibodies may be involved in establishing PAH in SSc, albeit this has not yet been investigated in vivo [25].

Therefore, the activation of autoreactive B cells at the early disease stage, leading to the production of stimulating autoantibodies, might contribute to the abnormal activity of fibroblasts, as well as vascular damage and remodelling in later phases. However, none of these stimulatory autoantibodies has shown specificity for SSc, and their role in SSc pathophysiology remains a matter of debate.

### 2.2. Vasculopathy

Vascular abnormalities are among the earliest manifestations in SSc. The underlying molecular pathogenesis involves a complex interplay of cellular and molecular mechanisms contributing to endothelial dysfunction, vasculopathy, and subsequent clinical manifestations. ECs play a key role in maintaining vascular homeostasis [5]. In SSc, various insults (e.g., autoimmune responses and environmental factors) lead to ECs’ activation and injury. This process involves the dysregulation of adhesion molecules (selectins, intercellular adhesion molecule 1 ICAM-1, vascular cell adhesion molecule 1 VCAM-1, etc.), which promote leukocyte adhesion and migration into the vessel walls [26]. The overproduction of vasoactive mediators, such as ET-1, contributes to vasoconstriction, ECs proliferation and fibrosis. Elevated levels of ET-1 are associated with endothelial injury and dysfunction, resulting in widespread microvascular damage, abnormal angiogenesis, impaired vasoregulation, ischaemia, and tissue hypoxia [27,28]. Additionally, the loss of endothelial integrity triggers a cascade of events, including platelet activation, coagulation abnormalities, and increased vascular permeability. These contribute to the development of DUs, PAH, PHI, and SRC, which significantly impact the prognosis and quality of life of SSc patients [29].

### 2.3. Fibrosis

Progressive fibrosis, a hallmark of SSc, consists of the excessive deposition of collagen and extracellular matrix (ECM) components in various tissues, and fibroblasts are the main players in the fibrotic process. In SSc, cytokines, growth factors (e.g., TGF-β) and mechanical stress can trigger the activation of quiescent fibroblasts into myofibroblasts, which are characterised by increased contractility and enhanced collagen production, thus promoting tissue fibrosis [4]. Dysregulated signalling pathways, particularly those involving TGF-β, play a central role in promoting the excessive synthesis and deposition of aberrant ECM components—especially collagen—which disrupt the tissue architecture, leading to fibrosis in the skin and internal organs such as the lungs, heart, gastrointestinal tract, and kidneys. This may result in organ dysfunction and failure, which are major determinants of mortality in SSc [30]. Immune dysregulation promotes fibroblast activation and collagen synthesis through the infiltration of immune cells (e.g., T lymphocytes) and the release of pro-inflammatory cytokines. Autoantibodies targeting specific cellular components further exacerbate the pro-inflammatory state, contributing to fibrosis and tissue damage [31].

## 3. Immunological Response and Production of Autoantibodies Targeting GPCRs

GPCRs represent the largest super-family of integral membrane proteins found in humans, boasting over 1000 distinct members. The fundamental structural component within a GPCR is the seven-transmembrane receptor domain, which employs GTP-binding proteins to facilitate signal transduction [32]. GPCRs are broadly detectable in non-immune cells like ECs and fibroblasts, as well as innate immune cells such as neutrophils, monocytes, macrophages, and dendritic cells. Additionally, they are also found in adaptive immune cells like lymphocytes, and they may be overexpressed in selected tissues [33]. The presence of functional antibodies against various GPCRs in human serum has been found to have agonistic or antagonistic activity, and these interactions play a role in regulating immune responses and physiological processes [34]. In the past, it was believed that antibodies against GPCRs always led to autoimmune diseases. However, there is now growing recognition of the complex role of these antibodies in controlling autoimmunity and their protective effects against some immune-mediated diseases, such as psoriasis and type 1 diabetes [35].

Antibodies against GPCRs, such as AT1R, ETAR, and C-X-C motif chemokine receptors (CXCR3 and CXCR4), can attract immune cells that express the corresponding receptors or be attracted by those cells, similarly to endogenous ligands and their receptors [34,36,37]. This interaction is important for immune cell homeostasis between the blood and tissues. The balance between the serum levels of antibodies against GPCRs and GPCR expression levels on immune or tissue-resident cells regulates cell migration towards the tissues, preventing a systemic immune response to a local injury.

## 4. SSc Clinical Manifestations Associated with GPCR-Targeting Autoantibodies

Antibodies targeting GPCRs have emerged as potentially crucial players in the pathogenesis of SSc, with significant clinical implications in both vascular and gastro-intestinal manifestations within affected patients. The main clinical associations are summarised in Table 1 and depicted in Figure 1. 

### 4.1. Associations between Antibodies Targeting GPCRs and Vasculopathy

Both immune and non-immune cell types, including vascular smooth muscle cells, ECs, and fibroblasts, express ETAR and AT1R. These receptors are activated by ET1 and angiotensin II (Ang II), respectively [38]; ET1 and Ang II are potent vasoconstrictors that trigger actin polymerisation, thereby regulating cytoskeletal remodelling in vascular smooth muscle cells and immune cells, hence their pivotal role in controlling blood pressure and facilitating immune cell trafficking [27,39]. The immuno-pathological mechanisms underlying SSc—vasoconstriction, as well as pro-inflammatory, proliferative, and profibrotic effects—are closely associated with the molecular events mediated by Ang II and ET1 through AT1R and ETAR, respectively. Therefore, the hypothesis positing that agonistic autoantibodies targeting these vascular receptors may contribute to the pathogenesis of SSc has been extensively investigated in recent years. Interestingly, these autoantibodies mimic ET-1 and Ang II and activate signal transduction pathways in non-immune and immune cells, as demonstrated by in vitro and in vivo studies [33]; they induce the production of IL-8 and adhesion molecules by ECs, stimulate T-cell chemotaxis and the secretion of IL-8 and C-C chemokine ligand 18 (CCL18) in monocytes, increase type I collagen deposition, and induce the expression of transforming TGF-β in human microvascular endothelial cells (HMECs), suggesting a potential involvement in fibrosis [40]. Elevated levels of antibodies against ATR1 and ETAR are found in approximately 85% of SSc patients [16]. These antibodies directly contribute to the initiation of vascular inflammation and fibrosis in vitro and in vivo by activating ECs, fibroblasts, and neutrophils, thus contributing to key pathological manifestation in SSc [40]. Unsurprisingly, their presence in the sera of SSc patients directly correlates with major disease manifestations [16].

Additional players have been recently investigated as potential contributors to SSc-related vascular dysfunction. Protease-Activated Receptor 1 (PAR1) is a GPCR that interacts with multiple G protein subfamilies and their linked signalling pathways to regulate a wide range of pathophysiological processes [41]. PAR-1 can be found in different cell types, like ECs and smooth muscle cells, and plays a crucial role in the regulation of endothelial barrier function and the production of pro-inflammatory cytokines (e.g., IL-6) in various inflammatory conditions, including autoimmune diseases [41].

**Table 1 ijms-25-02299-t001:** Overview of autoantibodies targeting GPCRs and their clinical association in SSc.

Autoantibodies against GPCRs	Gene	Stimulating/ Inhibiting	Patho-Physiological Effects	SSc-Related Clinical Manifestation	References
**Angiotensin II type 1 receptor (AT1R)**	AGTR1	Stimulating	Vasoconstriction Expression of adhesion molecules (VCAM-1) by HMECs Production of pro-inflammatory cytokines (IL-6, IL-8, TNF, CCL18) by HMEC and leukocytes Collagen production by fibroblasts	Pulmonary Arterial Hypertension	Riemekasten et al. [16] Becker et al. [42]
				Scleroderma Renal Crisis	Hegner et al. [43]
**Endothelin-1 type A receptor (ETAR)**	EDNRA	Stimulating	Vasoconstriction Expression of adhesion molecules (VCAM-1) by HMECs Production of pro-inflammatory cytokines (IL-6, IL-8, TNF, CCL18) by HMEC and leukocytes Collagen production by fibroblasts	Digital Ulcers	Avouac et al. [44]
				Pulmonary Arterial Hypertension	Riemekasten et al. [16] Becker et al. [42]
				Scleroderma Renal Crisis	Hegner et al. [43]
**Endothelin-1 type B receptor (ETBR)**	EDNRB	Stimulating	Same as anti-ETAR	Pulmonary Arterial Hypertension	Tabeling et al. [45]
**C-X-C motif** **chemokine receptor type 3 (CXCR3) and type 4 (CXCR4)**	CXCR3 CXCR4	Stimulating	Increasing gene expression of adhesion molecules, pro-inflammatory cytokines, and proteins of ECM	Pulmonary Fibrosis	Weigold et al. [36]
**Protease-activated receptor-1 (PAR-1)**	F2R	Stimulating	Production of pro-inflammatory cytokines (IL-6) by ECs	Scleroderma Renal Crisis	Simon et al. [46]
**Muscarinic-3 Acetylcholine Receptor (M3R)**	CHRM3	Inhibiting	Impairment of cholinergic neurotransmission into the myenteric plexus of visceral smooth muscle cells	Gastrointestinal Involvement (Dysmotility)	Kawaguchi et al. [47]

CCL18: C-C chemokine ligand 18; ECs: endothelial cells; ECM: extracellular matrix; GPCRs: G protein-coupled receptors; HMEC: human microvascular endothelial cells; IL-6: interleukin-6; IL-8: interleukin-8; SSc: systemic sclerosis; TNF: tumor necrosis factor; VCAM-1: vascular cell adhesion molecule 1.

#### 4.1.1. Digital Ulcers (DUs)

RP and ischaemic complications such as DUs are the main causes of disease-related morbidity in patients with SSc [48]. Severe vascular disease manifestations (e.g., DUs, PAH, and SRC) have been associated with higher titres of anti-AT1R and anti-ETAR antibodies, which also appear to predict SSc-related mortality [16]. A study by Avouac et al. focused in particular on the significance of anti-ETAR antibodies in predicting the development of new DUs in patients with SSc [44]. Anti-ETAR antibodies were found to be more effective than other biomarkers in predicting new DUs, with a stronger predictive value vs. anti-AT1R antibodies. Anti-ETAR antibodies, combined with the active presence and/or history of DUs, were able to predict the new onset of DUs, thus potentially identifying patients who might benefit from early specific management or preventive strategies. They could also play a role in the development of scleroderma ulcers by triggering inflammatory and fibrotic events, indicating that treatments targeting these antibodies may open new therapeutic avenues in the future [49].

#### 4.1.2. Pulmonary Arterial Hypertension (PAH)

PAH is a life-threatening complication of connective tissue diseases (CTDs) characterised by increased pulmonary arterial pressure and pulmonary vascular resistance, which may affect up to 7.8–12% of SSc patients over the course of the disease [50]. Both inflammation and vascular remodelling have emerged as significant pathogenetic mechanisms driving PAH onset and progression. Several studies to date have focused on the role of ET-1 and functional autoantibodies—particularly anti-ETAR and anti-AT1R—as biomarkers of CTD-PAH, and their serum levels are occasionally measured for research and clinical purposes in referral centres [51]. A study by Becker et al. reported that anti-AT1R and anti-ETAR antibodies were more frequently detectable in SSc-PAH or other CTDs vs. other forms of pulmonary hypertension and may serve as prognostic and predictive biomarkers in rheumatological patients [42]. A recent study also found higher titres of antibodies against endothelin type B receptor (anti-ETBR) in the sera of SSc-PAH patients vs. healthy controls [45]. Given the immunomodulatory role of ETBR, we could hypothesise that these autoantibodies may block its downstream signalling pathway, thus enhancing inflammation, but more studies are needed.

#### 4.1.3. Renal Involvement

Autopsy studies reveal occult renal pathology in 60–80% of SSc patients. Although an isolated reduction in glomerular filtration rate has been found to be common in patients with SSc, particularly in the diffuse cutaneous form of the disease, its prognostic significance remains unclear [52]. In contrast, the dramatic impact on prognosis has undoubtedly been documented in patients with SRC, a rare but potentially life-threatening complication of SSc characterised by malignant hypertension and oligo/anuric acute renal failure, which may affect up to 10–15% of patients [53]. SRC typically presents in patients with rapidly progressive diffuse cutaneous SSc (dcSSc) within the first 3–5 years after the onset of a non-Raynaud sign or symptom [54]. Recent evidence suggests the presence of anti-PAR-1 antibodies in SSc patients experiencing SRC [46]. Studies show that HMECs stimulated by SSc-IgG-related antibodies activate PAR-1, thus initiating a signalling sequence that involves the phosphatidylinositol-3-kinase/mammalian target of the rapamycin/extracellular signal-regulated kinases 1/2 (PI3K/mTOR/ERK1/2) pathway and the activator protein 1/c-FOS (AP-1/c-FOS) transcription factor. This sequence activates the IL-6 promoter and increases IL-6 secretion, which is influenced by both the duration of exposure and the antibody concentration [46].

As mentioned earlier, high levels of autoantibodies directed against AT1R and ETAR were associated with an increased risk of vascular complications, including SRC [16]. A recent study found evidence of pathophysiological mechanisms involving agonistic anti-AT1R and anti-ETAR antibodies in SRC, not only via the activation of ECs but also by increasing the contractility of renal resistance interlobar arteries, as well as hypersensitization to its natural ligands Ang II and ET-1, and the mediation of crosstalk between the two receptors [43]. Anti-AT1R antibodies increase the sensitivity of AT1R to its natural ligand Ang II and may play a role in the development of “renal Raynaud’s phenomenon”, which has been suggested to contribute to SRC onset [39].

#### 4.1.4. Primary Heart Involvement (PHI)

The role of anti-GPCR autoantibodies in the pathogenesis of PHI has not been investigated yet. PHI is a frequent and underdiagnosed complication of SSc [55,56,57,58] and is associated with poor outcomes [59,60]. Two different mechanisms, potentially triggered by anti-GPCR autoantibodies and synergistically associated with myocardial fibrosis [61], are pivotal in the pathogenesis of PHI: microvascular dysfunction [8,9,62,63,64,65] and myocardial inflammation [66,67]. Regarding inflammation, anti-ETAR and anti-AT1R autoantibodies are known to elicit specific intracellular inflammatory pathways [38,68], which could contribute to myocardial inflammation. With regard to microvascular dysfunction, the potential association with anti-GPCR autoantibodies is even more intriguing, and while the general pro-vasoconstrictive effects of these autoantibodies have long been known [33,38], their specific role in heart disease [69] and in coronary microcirculation has only recently been investigated. Interestingly, anti-ETAR autoantibodies have been found to be associated with coronary microvascular obstruction after acute myocardial infarction [70], a phenomenon in which coronary microcirculation is fully dysfunctional and prevents myocardial reperfusion. The association between anti-GPCR autoantibodies and important determinants of PHI (such as myocardial fibrosis, myocardial inflammation, and coronary microvascular dysfunction) suggests that anti-GPCR autoantibodies might be involved in the pathogenesis of SSc-PHI; however, further studies are needed to confirm these hypotheses.

### 4.2. Associations between Antibodies Targeting GPCRs and Interstitial Lung Disease (ILD)

ILD is a frequent organ manifestation in SSc, whose disease course varies widely, ranging from mild and stable disease to severe and rapidly progressing [71]. The pathogenesis of SSc-ILD remains complex and not fully elucidated. Like other forms of ILD, SSc-ILD consists of fibrosis with varying degrees of inflammation. SSc-related fibrosis is characterised by activation of both the innate and adaptive immune system, resulting in fibroblast activation and myofibroblasts producing excessive ECM [72]. A study by Weigold et al. investigated the presence and levels of autoantibodies against CXCR3 and CXCR4 in SSc patients, exploring their impact on fibrosis, a key feature in SSc [36]. CXCR3 and CXCR4 are GPCRs that mediate the migration and homing of lymphocytes, endothelial progenitor cells, and stem cells [73,74]. Similar to the correlation between anti-AT1R and anti-ETAR autoantibodies, anti-CXCR3 and anti-CXCR4 autoantibodies levels also showed a strong correlation with each other. Patients affected by SSc-ILD and diffuse cutaneous involvement exhibited higher median titres of anti-CXCR3 and anti-CXCR4 antibodies vs. limited cutaneous SSc (lcSSc) and healthy controls. Analyses of antibody levels in relation to lung involvement yielded unexpected results. Elevated antibody titres correlated negatively with pulmonary function parameters—i.e., more severe lung function impairment. However, upon analysing sera from SSc patients with a minimum 3-year follow-up, decreased levels of anti-CXCR3/4 antibodies seemed to be predictive of worsening of pulmonary function parameters [36]. Thus, the clinical significance of anti-CXCR3/4 levels remains uncertain, representing markers for both more severe yet more stable lung disease. These findings (in light of the profibrotic effect of CXCR3 and CXCR4) need further investigation to clarify the complex activation/inhibition effects resulting from the interaction between these chemokine receptors and their corresponding antibodies.

### 4.3. Associations between Antibodies Targeting GPCRs and Gastrointestinal Involvement

Gastrointestinal (GI) involvement is very common in SSc patients and virtually any part of the GI tract can be affected by the disease [75,76]. Nonetheless, the pathogenesis of such manifestations remains largely unknown: it has been proposed that GI involvement in SSc happens in three steps, namely, neural dysfunction, smooth muscle atrophy and fibrosis [77]. However, more recent findings indicate that, especially at the early stages of the disease, severe GI manifestations could be the result of an immune-mediated process targeting the enteric nervous system (ENS), causing gastrointestinal dysmotility. The ENS controls the contractile activity of the GI tract predominantly via cholinergic neurotransmission: intrinsic neurons of the myenteric plexus release acetylcholine (ACh), which promotes peristaltic movements via the muscarinic receptors M1-M5, located on smooth muscle cells. A study by Kawaguchi et al. found significantly higher titres of antagonistic autoantibodies targeting the muscarinic-3 acetylcholine receptor (M3R) in the sera of SSc patients with severe GI involvement (i.e., malabsorption, pseudo-obstruction and/or need for parenteral hyperalimentation) within 2 years of SSc diagnosis vs. SSc patients without early GI involvement [47]. This finding was later confirmed in other studies [78,79]. It is worth noting that a more recent study found autoantibodies against α3 and β4 subunits of nicotinic cholinergic receptor (anti-gAChR), which mediate neurotransmission in autonomic ganglia, in SSc patients with GI involvement [80], indicating that an ACh-mediated autonomic neuropathy may play an additional role in the complex pathogenesis of SSc-related gastrointestinal dysmotility.
Figure 1Main correlations between functional antibodies targeting G-protein-coupled receptors (GPCRs) and distinct clinical manifestations in SSc. Autoantibodies against AT1R, ETAR/ETBR, CXCR3/4, and PAR1 amplify constitutive inflammatory responses via their signalling pathways. The release of mediators from smooth muscle cells and the accumulation of pro-inflammatory cytokines further contribute to severe vascular dysfunction (vasculopathy). Autoantibodies against M3R disrupt the neurotransmission required for gastrointestinal motility. AT1R, angiotensin receptor type 1; CXCR3/4, C-X-C motif chemokine receptor type 3/4; ETAR, endothelin receptor type A; ETBR, endothelin receptor type B; M3R, muscarinic acetylcholine receptor 3; PAR1, protease-activated receptor 1. Created with BioRender.com.
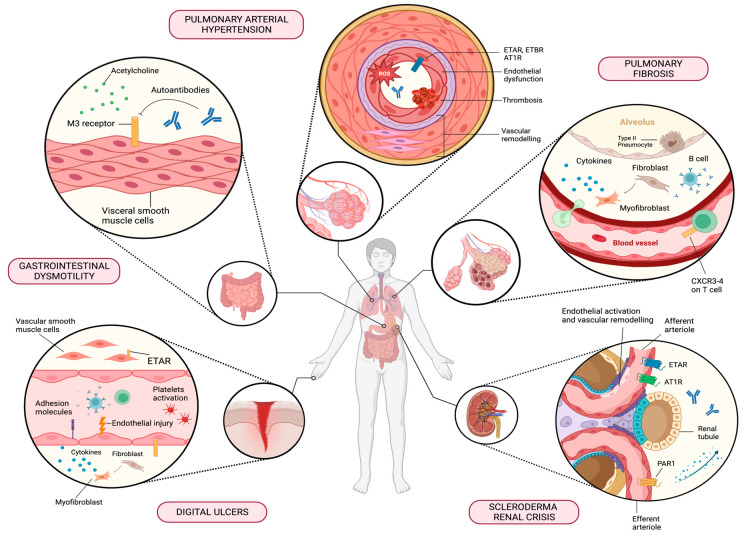


## 5. Potential Therapeutic Implications

SSc is characterised by significant heterogeneity, ranging from mild and stable to life-threatening forms. So far, the therapeutic approach to SSc has been based on the clinical manifestations of the disease, targeting the main pathogenetic mechanisms: autoimmunity, vasculopathy or, more recently, fibrosis [81,82]. Although new drugs have been approved for the treatment of SSc in recent years [82,83], the prognosis remains poor and SSc carries a high mortality risk [84]. Hence, considerable efforts have been made to identify novel therapeutic targets in SSc; in this regard, antagonising GPCRs and their specific autoantibodies could be a promising strategy [85]. However, there is scant evidence in support of the use of anti-GPCR agents in systemic autoimmune diseases, bearing in mind that GPCRs can activate a plethora of pathways, and this may reduce drug efficacy or induce undesirable effects. In this section, we summarise the currently available data on the potential use of such drugs in SSc according to their mechanism of action.

### 5.1. GPCRs Blockade

To date, only endothelin receptor antagonists (ERA), which block both ETAR and ETBR, have been successfully used to treat vasculopathic manifestations in SSc, such as PAH and DUs [86,87]. In SRC, the inhibition of angiotensin II-mediated vasoconstriction plays a crucial role in preserving renal function, with ACE-inhibitors as a first-line treatment [88]. There is less robust evidence regarding the role of endothelin receptor blockade in SSc-related renal vasculopathy. A small pilot study (BIRD-1) evaluated the benefits of combination therapy with bosentan and ACE-inhibitors vs. ACE-inhibitor monotherapy over 6 months in patients with SRC, and found a similar overall mortality rate in both cohorts. The overall frequency of dialysis was lower, and the renal function was better in the former, though the differences were not statistically significant, likely due to the small sample size [89]. More recently, a randomised placebo-controlled phase II trial on the highly selective ETAR blocker zibotentan showed a significant improvement in renal function at 52 weeks in SSc patients with chronic kidney disease in the zibotentan arm, without significant differences between the two groups regarding the expression of serum VCAM1—a candidate biomarker of SRC [90]. These findings appear to suggest a beneficial role for ETAR blockade in the long-term improvement of SSc-related renal vasculopathy rather than in an acute setting, but additional data are needed.

More recently, another potential therapeutic avenue for SSc was found to be the pharmacological inhibition of fibrosis via lysophospholipids (LPs) signalling pathways. Lysophosphatidic acid (LPA), a bioactive lipid mediator derived from membrane phospholipids in response to inflammation or cell injury, binds to its specific GPCR, LPA receptor (LPA1), activating various intracellular pathways. Its signalling is associated with skin and pulmonary fibrosis, suggesting a potential therapeutic target in fibrotic diseases, namely SSc [91,92]. A novel oral antagonist of the LPA1 receptor, SAR100842, was recently tested in SSc. A double-blind, randomised, placebo-controlled study was conducted in 17 patients with early (disease duration < 36 months) dcSSc. At week 8, the drug was well tolerated and a greater reduction in the modified Rodnan Skin Score (mRSS) was observed in the treatment arm vs. placebo, though the difference was not statistically significant. An LPA-related gene analysis in skin biopsies confirmed LPA1 target engagement [93].

### 5.2. Neutralisation or Elimination of Pathogenic Antibodies

In several autoimmune diseases, autoantibodies and immune complexes are directly involved in the pathogenesis and strongly correlate with clinical manifestations. Thus, the removal or neutralisation of such antibodies is a successful therapeutic strategy in many conditions [94]. In fact, immunoadsorption and plasma exchange have been tested in SSc with promising results. However, due to the low quality of the data, the latest American Society for Apheresis (ASFA) guidelines do not strongly support the use of therapeutic apheresis in SSc (category III) [95]. To date, available clinical data appear to suggest that therapeutic plasma exchange may improve skin involvement, SRC with signs of microangiopathy and vasculopathic manifestations [88,96,97], but further studies are needed. It bears noting that SSc-related antibodies (including antibodies targeting GPCRs such as anti-ETAR and anti-AT1R) are quite refractory to conventional immunoadsorption, as their levels are rapidly restored once treatment is discontinued [98]. To overcome this concern, an improved immunoadsorption technique was developed by using nucleic acid aptamers or so-called “chemical antibodies”, which are single-stranded RNA or DNA oligonucleotide sequences that bind to a specific target molecule in its native conformation with high affinity. Aptamer BC007 has been shown to decrease anti-GPCRs activity in vitro and in animal models [99]. Given the beneficial characteristics of aptamers in the neutralisation of antibodies, this approach could offer a promising new therapeutic strategy for the treatment of SSc.

Intravenous immunoglobulins (IVIGs) are a blood product obtained by the fractionation of plasma pooled from many donors and mainly comprise the IgG isotype. Although their mechanism of action is more complex, the rationale for their use lies mostly in their capacity to bind Fc fragments of pathogenic autoantibodies, thus preventing their interaction with the antigen and favouring their elimination [100]. Hence, IVIGs are widely used in the treatment of several autoimmune conditions and appear to improve several clinical manifestations of SSc [101,102,103,104]. Nevertheless, IVIGs are not yet indicated for the treatment of SSc, due to the lack of high-quality evidence. Recent retrospective data from a multicentric SSc cohort study (78 patients) found that IVIG treatment yielded benefits for myositis, skin, and gastrointestinal involvement [105]. Regarding gastrointestinal involvement, it is worth noting that the efficacy of IVIG treatment was assessed in light of the recent detection of anti-M3R antibodies. Ex vivo studies demonstrated that pooled human IgG were able to reverse the cholinergic dysfunction associated with SSc-related gastrointestinal disease by directly neutralising functional anti-M3R antibodies [79,106]. A recent case report highlighted the efficacy of IVIG treatment in two SSc patients with anti-M3R positivity and upper gastrointestinal tract involvement [107], but further in vivo studies are needed to clarify the predictive value of anti-M3R antibodies for treatment response.

Autologous haematopoietic stem cell transplantation (HSCT) is another therapeutic option for the management of severe SSc manifestations. Given the relatively high periprocedural risk, the current guidelines recommend the use of HSCT only in highly selected patients with rapidly progressive SSc at risk of organ failure [81]. The effectiveness of this method is based on the elimination of autoreactive lymphocytes by using a high dose of immunosuppressants and the subsequent restoration of the immune response via the transplantation of haemopoietic stem cells [108]. However, the effect of HSCT on antibodies targeting GPCRs has not been fully clarified yet. A recent study found that the titre of stimulatory anti-AT1R antibodies decreased after HSCT in patients with SSc, whereas their reactivity was not influenced by the treatment [109], suggesting that HSCT may not be effective in improving overall anti-GPCR-mediated manifestations.

### 5.3. Inhibition of GPCRs Downstream G-Protein-Driven Pathways: Biased Agonism

GPCRs are highly dynamic proteins that can adopt multiple ligand-induced conformations for the recruitment of their distinct signalling partners to deliver various cellular functions. It has been demonstrated that GPCRs can selectively activate intracellular signal transduction via G proteins or skew signalling towards other intracellular pathways, such as ß-arrestin [110]. The current literature shows that G-protein- and ß-arrestin-mediated pathways are not activated contemporaneously, though certain molecules, referred to as biased ligands, can selectively activate a subset of the receptor’s downstream signalling cascade by inducing conformational changes in the GPCR. This particular feature is known as “biased agonism” [111]. A better understanding of the mechanism of biased agonism has given rise to new challenges in drug discovery with the development of novel “biased” ligands, which preferentially induce ß-arrestin and consequently uncouple GPCRs from G proteins, resulting in improved therapeutic and safety profiles. One of the most studied GPCRs with respect to biased agonism and its therapeutic implications, mostly for cardiovascular morbidities, is AT1R. In animal models, a ß-arrestin-biased AT1R ligand—TRV120023—was able to improve cardiac contractility and protect cardiomyocytes from ischaemia-reperfusion or mechanical damage, with a reduction in cellular apoptosis, when compared with losartan, an unbiased AT1R blocker—which thus appears to inhibit both G-protein- and ß-arrestin-mediated pathways [112]. Another ß-arrestin-biased AT1R agonist, TRV120027, was shown to increase cardiac contractility in vitro, and to decrease blood pressure [113] and improve aortic aneurism-related mortality in mice [114]. However, in two recent clinical trials, TRV120027 did not improve clinical status in patients with acute heart failure and COVID-19 [115,116]. The exact reason for the failure of such a therapeutic approach in vivo remains unknown, but it has been hypothesised that it could be due to the functional differences between the two ß-arrestin proteins (i.e., cardiotoxic ß-arrestin-1 vs. cardioprotective ß-arrestin-2), and particularly to the predominant expression of ß-arrestin-1 in human cardiomyocytes [12,117].

Data pertaining to other GPCRs are scarce. Concerning ETAR, a study by Xiong et al. demonstrated that in animal models, ß-arrestin-1 and 2 are not responsible for ET-1-mediated vasoconstriction and may not play a role in ET-1 receptor desensitisation, thus suggesting a more limited role for biased selectivity in antagonising the detrimental effect of an enhanced ETAR downstream pathway [118]. Other studies have demonstrated that the binding of activated protein C (APC) to the endothelial protein C receptor Induces the activation of a ß-arrestin-2 PAR1-mediated intracellular transduction pathway, with protective effects against apoptosis in Ecs, unlike the effect of thrombin [119,120,121]. Based on this finding, Roy et al. pre-treated Ecs with catalytically inactive protein C (PC-S195A) in an in vivo inflammatory model, confirming a positive cytoprotective effect against thrombin in treated cells [121]. Biased agonism has also been demonstrated for CXCR3 and 4, as well as for M3R, but no therapeutic strategies using such a mechanism have been developed so far [122,123,124].

To date, none of the aforementioned strategies have been tested in SSc models. Therefore, further research is needed to ascertain the possible benefits of biased agonism as a novel treatment option in patients with SSc.

## 6. Conclusions

Despite recent advances in the understanding of its pathogenesis, SSc remains a challenging disease with a wide spectrum of severity and a high mortality rate. There is growing evidence that abnormal levels of autoantibodies against GPCRs, as well as their interaction with various ligands, may influence the development and progression of the disease. Notably, GPCR-targeting autoantibodies have gained prominence for their potential role in initiating vasculopathy and maintaining fibrotic processes, further increasing morbidity and mortality risk in SSc patients. Beyond diagnostic purposes, the precise characterisation of the clinical manifestations associated with these antibodies may not only shed light on disease progression but could also usher in novel targeted therapeutic interventions and improve patient outcomes.

Several efforts have been carried out to understand the role of GPCR-targeting functional autoimmunity, yet its impact on haematopoiesis remains to be investigated. Ongoing studies aim to identify novel GPCRs targeted by autoantibodies in SSc, providing insights into the development of diagnostic and prognostic biomarkers to guide clinical management. Nonetheless, due to the complexity of GPCR pathways, larger prospective studies are needed to ascertain the value of all the aforementioned candidate biomarkers as potential tools in routine clinical practice to improve the organ-based management of complications and risk stratification.

## Data Availability

Not applicable.

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
