# Peer review of "Autoantibodies Targeting G-Protein-Coupled Receptors: Pathogenetic, Clinical and Therapeutic Implications in Systemic Sclerosis"

_ijms, 2024, doi:10.3390/ijms25042299_

Round 1

Reviewer 1 Report

Comments and Suggestions for Authors

Review should be revised accordingly.

Reviewer 2 Report

Comments and Suggestions for Authors

Comments to the Authors:

1.      English language (grammar, style! and punctuation) is in general of good quality, but I’ve found some errors that require minor revision, as listed below.

·        Page 2, line 94 and page 3, line 122: is “leucocyte” – should be “leukocyte” ;

·        Page 7, line 301: the word “resembling” is not suitable here; please use the more appropriate one

2.      In my opinion (based on published data) the statement used by the Authors “although SSc-specific autoantibodies do not appear to be directly pathogenic” (Abstract) as well as “SSc-specific antibodies (anti-topoisomerase I, anticentromere, anti-RNA polymerase III) do not appear to play a pathogenic role in the development of the disease (Introduction page 2, lines 51-53) and “Although most autoantibodies detected to date, such as anti-topoisomerase I antibodies or anti-centromere antibodies, have not yet been linked to the molecular pathogenesis of SSc” (page 2, lines 87-88) is repeated 2 x, and, more importantly, may be a misstatement -  please reconsider; see the article Senécal JL, Hoa S, Yang R, Koenig M. Pathogenic roles of autoantibodies in systemic sclerosis: Current understandings in pathogenesis. J Scleroderma Relat Disord. 2020 Jun;5(2):103-129. doi: 10.1177/2397198319870667.

3.      Page 5, Table 1: lack of “TNF” abbreviation under the Table

4.      Authors should provide the abbreviation only the first time they use the full name and additionally in Tables’ legend as well as (on the other hand) – provide the full name with the first use of the abbreviation and use the abbreviation consistently in the manuscript ! Please check carefully the entire manuscript and correct !!!;

·        e.g. page 7: “ILD”

·        endothelial cells – Authors do not use abbreviation i.e. ECs ! but full name in the whole manuscript – please change

5.      Authors should cite the original study from which data are provided, not a review paper that include such data.; e.g.  page 7, lines 296-300) “Elevated antibody titres correlated negatively with pulmonary function parameters – i.e. more severe lung function impairment. However, upon analysing sera from SSc patients with a minimum 3-year follow-up, decreased levels of anti-CXCR3/4 antibodies seemed to be predictive of worsening of pulmonary function parameters [67]. The reference 67 is a review paper by Distler et al., but the cited data comes from the original paper: Weigold F, Günther J, Pfeiffenberger M, Cabral-Marques O, Siegert E, Dragun D, Philippe A, Regensburger AK, Recke A, Yu X, Petersen F, Catar R, Biesen R, Hiepe F, Burmester GR, Heidecke H, Riemekasten G. Antibodies against chemokine receptors CXCR3 and CXCR4 predict progressive deterioration of lung function in patients with systemic sclerosis. Arthritis Res Ther. 2018 Mar 22;20(1):52. doi: 10.1186/s13075-018-1545-8.; PLEASE CORRECT!

6.      The layout / arrangement of the manuscript is somewhat chaotic – division into sections as well as their titles should be reconsidered and improved; e.g.

·        Sub-section 2.2. “Immunological response and production of autoantibodies targeting GPCRs” do not apply systemic sclerosis pathogenesis but is included in Section 2 titled “Hallmarks of Systemic Sclerosis (SSc) Pathogenesis”.

·        In view of above – I suggest Authors to change the title of Section 2 into the title from Sub-section 2.1 – “Overview of immune abnormalities, vasculopathy, and fibrosis in SSc” and consistently numbering of sub-section 2.1.1 into 2.1; 2.1.2 into 2.2.; 2.1.3. into 2.3.;  The Sub-section 2.2. should be a new section numbered 3; subsection 2.4 should be a new section numbered 4; subsection 3. should be a new section numbered 5;

·        Authors should change or complete titles of some of the subsection; e.g. current sub-section 3.1. has the title “ Associations between antibodies targeting GPCRs and vasculopathy” but sub-section 3.2. and 3.3. do not have the same info in the titles , i.e. concerning associations , but simple “Interstitial lung disease” and “Gastrointestinal involvement”, respectively  – please change appropriately !!!

·        In current sub-section 2.3. that apply to general mechanism linking PCR autoantibodies to vasculopathy and fibrosis in SSc - only data concerning mechanism of Ab against ATR1 and ETAR in SSc are included; data of general mechanisms of other reviewed GPCR Authors included in other sub-sections - concerning SSc Clinical Manifestations Associated with GPCR-Targeting Autoantibodies; It makes the manuscript chaotic – please use the uniform rule of arrangement – include general data of all reviewed GPCR mechanisms in SSc in one subsection (current subsection 2.3) or delate subsection 2.3. and include general mechanism of particular GPCR in further sub-sections considering associations between GPCRs and clinical manifestations of SSc.;

- e.g. in subsection 3.1.1. only clinical associations of anti-ETAR and anti-AT1R are provided (their general mechanisms are described in sub-section 2.3.) but in the sub-section 3.1.3. the part of the text (page 6, lines 240-244) “Protease-activated receptor 1 (PAR1) is a GPCR that interacts with multiple G-protein subfamilies and their linked signaling pathways to regulate a wide range of pathophysiological processes [48]. PAR-1 can be found in different cell types like endothelial and smooth muscle cells, and plays a crucial role in the regulation of endothelial barrier function and the production of pro-inflammatory cytokines (e.g., IL-6)” – applies to general mechanisms of these GPCRs not the clinical associations

·        Current Sub-section 4.1. has a title “Receptor blockade” – I suggest to use more precise title, i.e. “GPCRs receptor blockade”

7.      Authors should reconsider to left / delete the sub-section “Primary heart involvement” since this part has no data applying to SSc; in my opinion is too theoretic and should be deleted

8.      Data in current sub-section 4.3. are very valuable and interesting, however completely do not apply to SSc; would be suitable for review paper about general molecular mechanisms of GPCRs / anti-GPCRs but not for paper specified to SSc. Authors should consider to delete or shorten this sub-section, or to add data that better argue this sub-section in terms of SSc

Comments on the Quality of English Language

Comments on the Quality of English Language are included in my general comments and suggestions to Authors
